# Metamitron, a Photosynthetic Electron Transport Chain Inhibitor, Modulates the Photoprotective Mechanism of Apple Trees

**DOI:** 10.3390/plants10122803

**Published:** 2021-12-17

**Authors:** Yuval Tadmor, Amir Raz, Shira Reikin-Barak, Vivek Ambastha, Eli Shemesh, Yehoram Leshem, Omer Crane, Raphael A. Stern, Martin Goldway, Dan Tchernov, Oded Liran

**Affiliations:** 1Group of Agrophysics Studies, MIGAL—Galilee Research Institute, Kiryat Shemona, Upper Galilee 11016, Israel; yuvalt@migal.org.il; 2Group of Molecular Genetics in Agriculture, MIGAL—Galilee Research Institute, Kiryat Shemona, Upper Galilee 11016, Israel; amirr@migal.org.il (A.R.); Goldway@migal.org.il (M.G.); 3Faculty of Sciences and Technology, Tel-Hai Academic College, Kiryat-Shemona, Upper Galilee 12208, Israel; yoril@migal.org.il (Y.L.); raffi@migal.org.il (R.A.S.); 4Northern R&D, Kiryat Shemona, Upper Galilee 11016, Israel; shirrb@gmail.com (S.R.-B.); omerc@migal.org.il (O.C.); 5Group of Plant Development and Adaptation, MIGAL—Galilee Research Institute, Kiryat Shemona, Upper Galilee 11016, Israel; viveka@migal.org.il; 6Morris Kahn Marine Research Station, Department of Marine Biology, Leon H. Charney School of Marine Sciences, University of Haifa, Mt. Carmel, Haifa 3498838, Israel; eshemesh@univ.haifa.ac.il (E.S.); dtchernov@univ.haifa.ac.il (D.T.)

**Keywords:** apple, Golden Delicious, Top Red, fruitlet thinners, photosynthesis, light reactions, electron transport rate, photoprotective mechanism, state transitions, PSII repair cycle

## Abstract

Chemical thinning of apple fruitlets is an important practice as it reduces the natural fruit load and, therefore, increases the size of the final fruit for commercial markets. In apples, one chemical thinner used is Metamitron, which is sold as the commercial product Brevis^®^ (Adama, Ashdod, Israel). This thinner inhibits the electron transfer between Photosystem II and Quinone-b within light reactions of photosynthesis. In this study, we investigated the responses of two apple cultivars—Golden Delicious and Top Red—and photosynthetic light reactions after administration of Brevis^®^. The analysis revealed that the presence of the inhibitor affects both cultivars’ energetic status. The kinetics of the photoprotective mechanism’s sub-processes are attenuated in both cultivars, but this seems more severe in the Top Red cultivar. State transitions of the antenna and Photosystem II repair cycle are decreased substantially when the Metamitron concentration is above 0.6% in the Top Red cultivar but not in the Golden Delicious cultivar. These attenuations result from a biased absorbed energy distribution between photochemistry and photoprotection pathways in the two cultivars. We suggest that Metamitron inadvertently interacts with photoprotective mechanism-related enzymes in chloroplasts of apple tree leaves. Specifically, we hypothesize that it may interact with the kinases responsible for the induction of state transitions and the Photosystem II repair cycle.

## 1. Introduction

The chemical thinning of pip fruit trees is an essential practice in agriculture as it reduces the natural fruit load, which results in larger fruits and enhanced economic value [1]. The chemical thinner’s main purpose is to decrease sugar supply to the fruitlet and thus encourage it to drop [2,3]. One way to reduce carbohydrate synthesis is to use photosynthesis-related herbicides that inhibit the photosynthetic electron transport chain [2]. The most common photosynthetic herbicides include 3-(3,4-DiChlorophenyl)-1,1-diMethylUrea (DCMU, commercial name—Diuron) and 4-Amino-4,5-dihydro-3-methyl-6-phenyl-1,2,4-triazin-5-one (metamitron, commercial name—Brevis^®^). While DCMU has been substantially researched over the years [4], Metamitron’s mechanism of action and its correct application dosage for apple trees is still a subject of debate [5].

Most of the sunlight energy absorbed in the photosynthetic apparatus is directed towards photochemistry and Electron Transport Rate (ETR) [6]. However, not all the absorbed energy can be used. In fact, the physical properties of the chlorophyll a molecule present an inherent difficulty in the photosynthetic apparatus. When the chlorophyll holds energy for prolonged time without the ability to transfer it forward, the chlorophyll changes its chemical energetic state and may interact with molecular oxygen, a reaction that results in Reactive Oxygen Species (ROS) [7]. ROS will damage the photosynthetic apparatus, reducing its capability to synthesize sugars. Then, the photosynthetic apparatus will commence several energy dissipation processes in order to mitigate the probability to induce ROS [8].

These processes are termed Non Photochemical Quenching (NPQ) [9], and are activated in response to the duration of the stress. Essentially, there are three major processes within NPQ that are activated at different time scales:Energy related (qE, several seconds) [10], responsible for the epoxidation of special carotenoids (Xanthophylls) located on the Light Harvesting Complexes (LHC, photosynthetic antenna). Upon epoxidation, these Xanthophylls change their chemical activity from light harvesting to energy dissipation and thus prevent light energy from reaching the reaction centers of PhotoSystem II (PSII).State transition related (qT, minutes) [11], responsible for transition of the LHCII between the two central photosynthetic complexes—PSII and Photosystem I (PSI). This results in a different allocation of absorbed energy between the two photosystems and reduces the damage to PSII.Photoinhibition dependent (qI, hours) [12], with prolonged stress, ROS will eventually accumulate in the vicinity of PSII and destroy the D1 subunit an important component of the reaction center [13]. The destruction of D1 results in a non-active PSII. The damaged D1 are marked for replacement via a regulated process that replaces the damaged unit with a de novo synthesized copy [14].

At the molecular level, two out of the three NPQ processes involve phosphorylation kinases of the State TraNsition (STN) family [15,16]. The first process is LHCII transfer between the reaction centers. Upon excess illumination, the trans-thylakoid membrane potential increases, and this induces STN7 activity [17]. Then, this kinase phosphorylates LHCII which results in the disconnection from PSII supercomplexes [15]. In the second process, the PSII repair cycle is comprised of a distinct set of steps which enables a rapid replacement of damaged D1 subunit [18]. PSII supercomplexes with damaged D1 are designated for a repair through phosphorylation via the STN8 pathway [16]. Then, a set of serine proteases cleave the transmembrane Psb-a protein (D1) at specific locations, thus exposing it to the repair complex [13].

When an ETR herbicide is applied, the absorbed light energy cannot be utilized in the light reactions. This event increases the probability of ROS accumulation, and damage to PSII. In fact, using such specific PSII inhibitors signals to the photosynthetic apparatus that the electron transport chain is reduced downstream PSII. This state mimics photosynthetic stress which, in turn, induces the photoprotective mechanisms [19,20]. Characterizing the response of the photosynthesis physiology in the presence of the inhibitor can then be performed with chlorophyll fluorescence methods [21,22]. These incorporate the fact that the photochemistry is linked to fluorescence emission via the interplay between the three fates of energy utilization within the apparatus—photochemistry, heat dissipation and fluorescence emission [6]. The most common examination includes a light response curve that examines the magnitude of the electron transport rate against a gradient of light intensities [23]. This curve includes several key parameters that explain the response of the apparatus to light: a. Light Use Efficiency (LUE)—the initial linear slope receives units of electrons excited in photochemistry to the amount of photons absorbed in the process [24,25]; b. The concave part of the curve marks the transition between photochemical to non-photochemical energy quenching, where absorbed energy is also being dissipated by other photoprotective mechanisms [26] to photochemistry [27]. Additional examinations using chlorophyll fluorescence include, but are not limited to, electron transitions within the PSII complex itself (OJIP test—[28]), and determining energy distribution between photochemistry and non-photochemical processes [29].

In this study, we report that Metamitron, an electron transport chain inhibitor, obtains an additional activity on top of its primary role. It interacts with the photoprotective mechanism of apple trees. We present evidence of two additional activities: it promotes a forced state transition of LHCII to PSI site in Top Red (TR) but not Golden Delicious (GD) cultivars; and interferes with the PSII repair cycle in both cultivars. These actions force the apparatus off balance which results in an unexpected photosynthetic physiology after Metamitron application.

## 2. Results

### 2.1. Parameterization of Light Response Curves of Two Cultivars of Apple Trees Leaves

Light response curves of PSII activity were performed with a background of Brevis^®^ in order to characterize the photosynthetic response to a block at the PSII site. Light response curves of GD (Figure 1A) exhibited a logarithmic response for the control group with the ETR reaching up to 60 μmol e^−^ m^−2^ s^−1^. The initial linear slope stretches until 100 μmol photons m^−2^ s^−1^, at which point the concave part of the curve starts to level out at 300 μmol photons m^−2^ s^−1^ towards its maximum activity. An increase in the inhibitor concentration results in a gradual decrease of the curve’s plateau. At 0.64% inhibitor concentration, the curve reaches a local maximum at 300 μmol photons m^−2^ s^−1^ and rapidly declines to zero, and at a concentration of 1.28% of inhibitor, there is no viable ETR at any light intensity. Light Use Efficiency (LUE), the initial linear slope of the light response curve (Figure 1B) of GD, shows a logarithmic decay towards the minimum LUE and then levels out at and above 0.32% inhibitor concentration. The LUE does not reach a zero rate, even at a concentration of 0.64–1.28% inhibitor. This means that at very low light intensities, there is still a minimum photosynthetic rate in the presence of the inhibitor. Effective antenna size (Figure 1C) presented a sigmoid behavior. The inflection point is found within the range of 0.08% to 0.16% inhibitor concentration, concomitant with a statistically significant decrease in the LUE data for this cultivar. Maximum photosynthetic activity (Figure 1D) presented a rate of logarithmic decay with a complete arrest in activity at the two highest inhibitor concentrations, as expected.

The TR (Figure 2A) ETR response to the Brevis^®^ inhibitor gradient was much more disorganized than that recorded for the GD cultivar. In the light response curve graph, there seems to be two groups of responses: below and above 0.16% inhibitor concentration. The light response curves for the control and the first group, presented a steeper increase in electron transport rate when compared with the same increase in the GD cultivar. Then, the activity reaches a maximum on the concave part of the curve at 500 μmol photons m^−2^ s^−1^ and not on the plateau. Finally, the curve starts to decline with greater light intensity, implying for development of photoinhibition of the TR cultivar at light intensities above 1000 μmol photons m^−2^ s^−1^. Additionally, the maximum activity reached was ~44 μmol e^−^ m^−2^ s^−1^ which is 75% of the maximum activity of the GD. With increase in inhibitor concentration, the curve magnitude decreases. The ETR curve does not reach zero in presence of even the highest inhibitor concentration. In fact, at 1.28% inhibitor concentration, the maximum rate reaches a higher activity than the 0.64% inhibitor concentration-related curve. When inspecting the LUE data (Figure 2B), the two groups are visible as expected, with the 0.16% inhibitor concentration corresponding statistically to the second group, that of the 0.32% inhibition concentration. There was no statistically significant difference between the LUE in the first group, as expected. Moreover, there was no gradual decrease in the efficiency. It seems as if above 0.16% inhibitor concentration, there is a “breaking point” and a forced decrease in the performance of PSII activity. When analyzing the effective antenna size (Figure 2C), there seems to be no organized decline with increasing concentration of the inhibitor, and on top of that, there is no statistical significance between the responses. This implies that the inhibitor interferes with some regulation pathway in control of the reactivity of PSII antenna to attenuation in light intensities. Finally, the maximum activity of TR (Figure 2D) is also divided into two activity groups, where each group has a similar appearance within themselves with no statistically significant difference. The only difference found between the two groups was the “breaking point” above 0.16% inhibitor concentration. Here, activity decreased to ~25% of the control sample activity. However, as seen for the ETR, this does not completely inhibit the activity of PSII.

### 2.2. Analysis of the Photoprotective Mechanism’s Response to the Metamitron Inhibitor

The yield of each of the three outcomes of light energy utilization by the photosynthetic apparatus was suggested by formulations in Kramer et al. (2004) (Figure 3). The yield of energy distribution was performed for the two apple cultivars, GD and TR, for the data recorded on the light response curve at three light intensities (Figure 3): a. 50 μmol photons m^−2^ s^−1^ (Figure 3A,B) which is found at the range of the initial linear slope of both of the cultivars and therefore relates to the LUE; b. 300 μmol photons m^−2^ s^−1^ (Figure 3C,D) which is found at the concave part of the curve and therefore relates to the antenna size attenuations of each cultivar; c. 1000 μmol photons m^−2^ s^−1^ (Figure 3E,F) which relates to the maximum ETR for each cultivar measured in this study.

The control group for GD at each of the three light intensities (Figure 3A,C,F, leftmost bar) presented a classic energy yield distribution, where at 50 μmol photons m^−2^ s^−1^, 60% of the energy was diverted to photochemistry (green fraction of the bar, Φ_II_). This fraction of quantum yield decreases with an increase in light intensity until 1000 μmol photons m^−2^ s^−1^, where the photochemistry yield reached below 20% (green fraction of the leftmost bar at Figure 3F). The quantum yield of regulated non-photochemical energy loss in PSII (Φ_NPQ_) increases with light intensity in the control up to 60% of the total absorbed energy at maximum light intensity (yellow fraction of the leftmost bar at Figure 3E). The fraction of energy lost by non-regulated processes in PSII (Φ_NO_) (blue fraction of the leftmost bar in the three panels, Figure 3A,C,E), remains constant at the 20% mark for each light intensity; this implies that under natural conditions, the ROS scavenging mechanism is efficient for this cultivar. With increasing concentration of the inhibitor there is a gradual decline in the photochemistry quantum yield for each of the three light intensities. The energy is diverted to the non-regulated loss of energy in PSII (Φ_NO_) and implies that the accumulation of triplet chlorophylls is expected (see the blue fraction of bars in each of the panels Figure 3A,C,E). The regulated non-photochemical energy loss in PSII (Φ_NPQ_) shows a statistically significant attenuation when passing the 0.08–0.16% of inhibitor concentration (note the yellow lettering in Figure 3A,C,E), as expected.

The energetic distribution between the three outcomes of light energy utilization in the case of the TR cultivar presented similar apparatus behavior from the control to that of the GD cultivar (Figure 3B,D,F). The energy lost by the non-regulated processes was slightly increased at 300 μmol photons m^−2^ s^−1^ when compared with that of the GD. The same phenomenon of energy lost by non-regulated NPQ, with increase in the concentration of the inhibitor, is seen for the TR when compared to the GD. However, one striking difference between the two cultivars is that there was no statistically significant difference between the yield of energy diverted to the regulated NPQ in any of the inhibitor concentration (note that there is almost no difference in magnitude of the yellow fraction of the bars in any of the three light intensities-Figure 3B,D,F). In addition, in TR and as seen before, there was no gradual decline in either of the three yields after an increase in inhibitor concentration. Instead, there is a “breaking point” between 0.08% to 0.16% inhibitor concentrations. The two groups below and above this threshold are significantly different from each other.

In order to explain the differences of allocated absorbed energy to the regulated non-photochemical energy loss in PSII (Φ_NPQ_) process between the two cultivars (yellow fraction of bars in Figure 3), we analyzed the relaxation kinetics of each of the photoprotective mechanisms of NPQ. That is, energy dependent quenching (qE), state transition dependent quenching (qT) and photoinhibition dependent quenching (qI) were examined against the background of the inhibitor gradient (Table 1 and Table 2 for GD and TR, respectively). The quenching values of each process were assessed over 10 days after application of the inhibitor, so as to track the response of the photoprotective mechanism along the natural decomposition rate of the inhibitor. The data presented on day 2 relates to the analysis performed in the previous steps shown. In the case of GD (Table 1), there was no difference between the dates for the level of energy dependent quenching (qE). This was expected, since there was no trans-thylakoid membrane potential created due to the presence of the inhibitor. There was a slight increasing trend each day, which may imply a generation of a proton potential gradient due to cyclic electron transport. The only statistically significant difference was found on the second day after application, where the energetic state was different between the 0.01% and 0.05% inhibitor concentrations. The state transition dependent quenching mechanism (Table 1, qT column) increased with inhibitor concentration, as expected due to the inhibition of electron transport downstream PSII. There was a statistically significant difference between the control and maximum inhibitor concentration, which implies that antennas are transferred from the PSII to the PSI site in order to reduce absorbed light energy reaching the PSII reaction center. Here, again the significant difference was present only two days after spraying, where the rest of the time there was no statistically significant difference. This implies a desensitizing effect of the inhibitor on the process. Eventually, the photoinhibition dependent quenching (Table 1, qI column) showed a decline in values with increasing concentration of the inhibitor. This increase was not statistically different on any of the dates except the second day after application. This is a surprising result, as the repair cycle does not increase its activity in the presence of the PSII inhibitor, which exposes the complex to an elevated level of absorbed energy. This increases the probability of ROS accumulation in the reaction center vicinity and therefore requires a replacement of damaged subunits. This also implies a desensitization of this mechanism by the inhibitor.

The TR cultivar presents a different, and to some extent opposite, behavior to the GD cultivar performance with respect to its photoprotective mechanisms (Table 2). The energetic dependent quenching of the TR (Table 2, qE column) increases with increasing concentration of the inhibitor. Although there is no statistical significance between the inhibitor concentration responses, it seems that the trans-thylakoid membrane potential increases, which implies for increased cyclic electron transport. This is because the inhibitor blocks electron transport downstream PSII, so the only source of electrons during the light period may be that of the PSI. The state transition dependent quenching magnitude (Table 2, qT column) decreases substantially with increasing concentration of the inhibitor for each time instance, implying a desensitization of the process with the application of the inhibitor. This, in fact, is surprising and requires an explanation for which of the two photosystems the antenna is found in this case. The only significant difference is found on the 5th day, but this trend repeats itself along the whole 10-day range. These results corroborate the decrease in magnitude of energy allocated to the regulated non-photochemical energy loss in PSII (Φ_NPQ_) process as analyzed in the previous step (Figure 3B). Finally, the photoinhibition dependent quenching of TR (Table 2, qI column) seems to relax with increasing concentration of the inhibitor up until the 5th day after application. Starting at the 7th day, there is a flip in the reactivity of this process, whereby the qI portion increases with increase in inhibitor concentration, probably due to the natural degradation of the inhibitor. The fact that there is a decrease in qI fraction during the first days after application, implies a desensitization of this process, because if the inhibitor concentration is higher, then more PSII complexes are blocked, and more energy is lost to the non-regulated dissipation mechanism (Figure 3, blue portion of graph in panels 3A, C, E). This behavior is not seen in the case of the GD as mentioned, and it also implies that a modulated repair mechanism of PSII in this cultivar may affect both the characteristics of electron transport rate and, by that, the quantum yield distribution profile as seen in the previous steps.

### 2.3. Metamitron Interferes with the Photoprotective Mechanism in Apple Trees

In view of the attenuated performance of both the state transitions and photoinhibition dependent quenching, a 77K fluorescence was analyzed. The 77K fluorescence spectrum presented a difference in fluorescent peak magnitudes of PSII (685 nm) and PSI (720–740 nm) in relation to the location of the antenna complex LHCII (Figure 4). GD that was exposed to a 0.6% inhibitor concentration presented a slight increase in the PSI peak over that of PSII (Figure 4A, green curve). This implies that there was an event of state transition after application of the inhibitor. Addition of DCMU, an inhibitor with similar mechanism of activity, did not show this phenomenon (Figure 4A, red curve). This implies that this phenotype is related only to Metamitron presence. There was almost no difference between these spectra and the control which was sprayed only with water (Figure 4A, blue curve). Contrary to this, the TR cultivar presented a very different fluorescence spectrum when Metamitron was applied (Figure 4B, green curve). Here, there was a very high magnitude in the fluorescence emitted from PSI—an increase in 15% fluorescence than PSII, which was not visible for GD and also not seen for TR when applied with DCMU (Figure 4B, compare green and red curves). There was a slight expansion of the PSI peak in the case of TR applied with DCMU, but this was not significant when compared to the control (Figure 4B, blue curve). These results imply that Metamitron encourages the transition state between the two photosystems when the apparatus was trying to avoid ROS accumulation in the vicinity of PSII, and the transfer of the antenna to PSI. These results corroborate the characteristics of the light response curve seen in Figure 2, where there was no statistical significance between the 0.64% and the control treatment. This suggests that the apparatus of at least the TR cultivar cannot regulate this process within the photoprotective mechanism when the Metamitron inhibitor is applied.

In view of the desensitization of the repair cycle of PSII in the TR cultivar, a Western blot analysis was performed on the D1 subunits with and without phosphorylation, to understand the dynamics between damaged D1 sent for degradation and the de novo synthesis of the new copy. In addition to these, OEC-33 scaffold subunit (Psb-O) was also checked, as it is very sensitive to the presence of ROS. In the case of GD (Figure 5A), non-phosphorylated Psb-a subunit (D1) shows a uniform appearance throughout all the concentrations of the inhibitor, and the same as the control (Figure 5A, upper panel). Phosphorylated Psb-a* showed strong bands in the control for both non-cleaved Psb-a* (35 kda) and cleaved product Psb-a* (25 kda). However, with increase in inhibitor concentration, GD’s cleaved Psb-a* decreased almost completely at 0.01% of the inhibitor (Figure 5A, middle panel), while the non-cleaved product was present at the same strength of the control only at the lower inhibitor concentrations. In the rest of the lanes, this subunit and cleavage product were very faint. The cleaved product of GD’s Psb-a* disappears completely above the concentration of 0.16%. The Psb-O subunit was also presented uniformly in the case of GD (Figure 5A, lower panel), despite very faint lines of cleaved product below the 35 kda band and between 0.01% and 0.08% inhibitor concentration.

The TR cultivar presented an arrested repair cycle of Psb-a (Figure 5B, upper panel). The control showed only a faint accumulation of Psb-a in contrast to the accumulation of this subunit which started at 0.01% inhibitor concentration; this increased substantially starting from the 0.16% inhibitor concentration. This implies that de novo copies of Psb-a are accumulated but may not be inserted into the thylakoid membrane. Phosphorylated Psb-a* presented a similar behavior between the control (where strong bands occur both for the intact and cleaved fragment). Starting from 0.08%, there was a complete disappearance of the cleaved product of Psb-a*. Instead, there was a mirror representation of the phosphorylated, non-cleaved Psb-a*. This implies that the non-cleaved phosphorylated subunit was not removed from the damaged PSII. This also explains the accumulation of the de novo synthesized copies recorded (compare Figure 5B upper and middle panels). Finally, the Psb-O subunit deteriorates starting at 0.08%, where a smear of the protein is visible (Figure 5B lower panel). This goes in line with the fact that damaged Psb-a* are not cleaved and thus are not being sent out of the PSII complex; this results in accumulation of ROS. Altogether, these results point to the fact that Metamitron interacts with at least two photoprotection mechanisms pathways and modulates their activity.

## 3. Discussion

This study provides evidence that Brevis^®^ (Adama, Israel), a Metamitron-based commercial chemical thinner, which is used as an inhibitor of the photosynthetic electron transport chain, interacts with the photoprotective mechanism of the photosynthetic apparatus of both apple cultivars Golden Delicious (GD) and Top Red (TR). Several photosynthetic physiological protocols were performed and provide proof that Metamitron suppresses the PSII repair cycle and forces a state transition of LHCII from PSII to PSI. These secondary activities related to the inhibitor, affect each of the photoprotective and light harvesting processes within the chloroplasts of TR. GD apparatus is also affected by the inhibitor, however to a lesser extent. The damaged D1 subunits of TR are accumulating within the thylakoid membranes in application of at least 0.08% inhibitor, and this results in accumulated ROS that disintegrates the Psb-O subunit (Figure 5B, lower panel). The Psb-O band profile in the gel is a sensitive marker for ROS accumulation within the chloroplasts, as was first suggested by Henmi et al. (2004) [30]. In their work, the researchers noted a degradation of Psb-O in response to very high light stress on the apparatus; they suggested that this degradation was caused by ROS. In the current study, there is a smear behavior seen in the gel of Psb-O when the inhibitor reaches above 0.04%, corroborating the damage by ROS in this cultivar. The accumulation of ROS at the vicinity of PSII encourages the LHCII to transfer to PSI in order to minimize energy flowing towards PSII (Figure 4B); this should decrease the accumulation of ROS [9]. In turn, the accumulation of ROS in the vicinity of PSII in the TR affects the response of this cultivar to increased light intensities on the background of the inhibitor. First, the relaxation of the repair cycle of TR is silenced as the inhibitor concentration increases (Table 2, photoinhibition dependent quenching (qI column)), and the qT state transition dependent quenching decreases as well, implying that after the forced transfer, the apparatus is stuck in this position (Table 2, qT column). This results in a non-statistically significant difference in antenna size in the case of the TR cultivar (Figure 2C). It also explains why there is no complete arrest of photosynthetic activity regardless of the very high concentration of the inhibitor (Figure 2A,D). This is probably since damaged PSII continues to work partially and are not being repaired. Rosa et al. (2021) show that at 0.2% administration of Metamitron in apple trees the ROS scavenging pathway is activated [4] This corroborates our findings that depend on a much higher concentration of Metamitron, and imply that ROS may indeed increase, which is verified on PSII complex molecular level. The modulation of photoprotective mechanisms was less indicative in the case of GD, although the fragmented D1 disappeared at the lowest concentration of the inhibitor (Figure 5A). In the GD case, the photoprotective mechanisms kept operating despite the attenuated performance of the repair cycle of damaged PSII. The gradual diversion of energy from regulated non-photochemical energy loss in PSII (Φ_NPQ_) to non-regulated loss of energy in PSII (Φ_NO_) is already documented and corroborated by the study of Klughammer and Schreiber (2008). In their paper, the researchers explain that with the closure of PSII and the decrease of the maximum fluorescence signal, most of the absorbed energy is diverted to non-regulated processes such as thermal decay, triplet chlorophyll, etc. Moreover, Wen et al. (2021) present evidence that in the model organism *Arabidopsis thaliana*, Deg-1 serine protease which is responsible for the cleavage of the phosphorylated Psb-a* is silenced, there is a marked decline in the electron transport rate [31]. Therefore, with an increase in inhibitor concentration there is a higher probability of the increase in ROS which interacts with the excited triplet chlorophyll within PSII. Finally, the light response curve of the GD is not harmed as expected in view of the intact Psb-a units (Figure 5A). GD presented a concave profile of carbon assimilation towards 1000 μmol photons m^−2^ s^−1^ [32], while other cultivars were shown to reach maximum photosynthetic activity already at 300 μmol photons m^−2^ s^−1^ [33]; both corroborate our results.

We therefore suggest that the commercial product Brevis^®^ interacts with at least two different photoprotective mechanisms of apple trees—the state transition and repair cycle of PSII. We suspect that this interaction may involve the STN kinases as both photoprotective mechanisms involved with these enzymes were affected by the presence of the inhibitor (Figure 6). Therefore, applying Metamitron affects not only electron transport rate out of PSII, but also modulates the photoprotection mechanism. In turn, this attenuates the capability of the apple trees to defend themselves from photosynthetic stress when applied with the Metamitron-based inhibitors.

The main limitation of this study is the fact that, except for the DCMU-positive control during the 77K fluorescence assay, we did not compare the commercial inhibitor activity to that of the pure substances Metamitron (4-Amino-4,5-dihydro-3-methyl-6-phenyl-1,2,4-triazin-5-one) and to DCMU (3-(3,4-dichlorophenyl)-1,1-dimethylurea). However, the analysis of an OJIP test (Appendix A) that examines the extent of inhibition within PSII on a gradient of inhibitor concentrations, was similar to the evidence presented by Abbaspoor et al. (2006). Moreover, there is a large body of evidence for this commercial inhibitor in relation to PSII inhibition [5,34], specifically in apple trees. In the case of DCMU, the non-regulated loss of energy in PSII (Φ_NO_) increases with inhibitor concentration and decreases the photochemical activity as shown in other studies [35,36]. An additional limitation is the fact that we did not corroborate the results of this study in outdoor conditions. The fact that the commercial inhibitor is metabolized with time in the plant tissues, together with the different activity profile between TR and GD, requires future study. Specifically, studying the attenuation over time of the inhibition under field conditions with the natural degradation of the inhibitor. Secondly, there was a discrepancy between the increase in inhibitor concentration and the decline in the regulated NPQ process, but there was a statistically significant increase in the qT portion in GD (Figure 3E compared to Table 1). We explain this discrepancy as such: while the photosynthetic apparatus cannot use the regulated NPQ processes, the antennas are still stuck in state-2 transition (on PSI, Figure 4B), so the relaxation for this process is prolonged. In addition, the other two quenching processes (Table 1, qE and qI) are not statistically significant. This corroborates the decline in the overall NPQ activity and was translated as a decrease in its quantum yield portion, as seen in Figure 3E for GD.

Lastly, it is noted that the greatest activity of this inhibitor under outdoor conditions occurred when the apple trees were shaded [37]. The evidence brought forth in this study provides an explanation why this is the case. This is because when the trees are shaded, there is less chance that the photosynthetic apparatus will push for induction of photoprotective pathways. Therefore, there would be less interaction between Brevis^®^ and the induced photoprotective enzymatic cascades. This lets the inhibitor perform its primary task, which is to inhibit the ETR within the photosynthetic apparatus and to reduce sugar synthesis.

Practical implication of the secondary activity of Metamitron will be to incorporate it into carbon balance models that predict the efficacy of fruitlet thinning depending on light and temperature [38]. The data gathered in this study should be considered to determine the optimal concentration of the inhibitor and its application timing. Under high irradiance, there is a higher chance that the Metamitron would interact with other components within the photosynthetic apparatus and attenuate the tree’s response to photosynthetic stress. At the crop level, it would be beneficial to research the response of other apple cultivars to Metamitron with respect to the photoprotective mechanisms within photosynthesis. This will expand our understanding of the different phenotypes of photoprotection within apple cultivars and contribute to our overall understanding of photosynthetic activity and photoprotection in this species. As Metamitron is being gradually introduced into other fruit species—e.g., citrus [39], peach [40], pears [41]—it will be beneficial to examine Metamitron reactivity with the photoprotective mechanisms of other fruit species as well. Finally, the reactivity of the photoprotective mechanisms to the inhibition of the ETR, as suggested in this study may be merely an extreme reaction of the NPQ pathways to the presence of the inhibitor, yet, both effects seen in the photoprotection are governed by the same family of kinases—STN8, which induces the serine protease responsible for cleavage of D1 and STN7 for phosphorylation of the LHC. It was implied that it interacts specifically with enzymes of this family. We, therefore, suggest to conduct a molecular investigation in order to establish a mechanistic model for Metamitron modulation of the respected cascades and to expand our understanding of its mode of action within the photosynthetic apparatus.

## 4. Materials and Methods

### 4.1. Plant Material & Chemicals Application

The experiment was conducted on two apple cultivars ‘Golden Delicious’ (GD) and ‘Top Red’ (TR), from the Fichman experimental farm in Israel (33.13199, 35.8059; *Malus domestica Bork.*). Branches were cut when fruitlet diameter was between 6–8 mm, corresponding to nine days after full bloom. The branches were acclimated for 24 h prior to experimentation in a growing room kept at 24.5 °C with 50 μmol photon m^−2^ s^−1^ cool white fluorescence light (OSRAM L36, W/840) and a 16:8 light:dark cycle. Each vase included three branches from three different trees of one cultivar in the experimental farms, for a total of 18 vases and 54 branches total for the two cultivars. Brevis^®^ (ADAMA, Ashdod, Israel) consists of a 15% active ingredient Metamitron (4-Amino-3-methyl-6-phenyl-1,2,4-triazin-5-one) that was dissolved in Double Distilled Water (DDW) according to the manufacturer instructions. A set of nine different concentrations were prepared in 1 L volumes: control (only DDW), 0.01%, 0.02%, 0.04%, 0.08%, 0.16%, 0.32%, 0.64% and 1.28%; these were sprayed in the morning after the 24 h acclimation in the growth chamber.

### 4.2. Chlorophyll a Fluorescence Measurements

A Pulse Amplitude Modulation (PAM)-based portable fluorometer (FlouroPen FP-100Max, Photon Systems Instruments, Brno, Czech Republic) with various pre-configured protocols was used. Experiments were carried out in green light by adhering green filter papers to headlight torches. The temperature was kept constant at 24.5 °C. The following protocols were used:OJIP [28] (Appendix A): The OJIP transient is a fast recording of the initial increase in fluorescence upon illumination of dark-adapted leaves. Leaves were adapted to the dark for 20 min. The OJIP initials stand for: O (F_0_) measured 50 µs after illumination starts, J—first intermediate step measured after 2 ms, I—second intermediate step measured after 30 ms, and P—Fm (maximum fluorescence) [42];Light response curve [23,43]: This protocol records maximum quantum yield in the dark and then effective quantum yield in series, each after an illumination period. In such a way, by gradually increasing the light, information regarding light use efficiency, characteristic light intensity and maximum activity of PSII can be extracted from the data. Maximum quantum yield is calculated as:
(1)$$\mathrm{Fv}/\mathrm{Fm}=\frac{F_{m}-F_{0}}{F_{m}}$$
where F_m_ and F_0_ are maximum and minimum fluorescence measured after dark adaptation time, respectively. Effective quantum yield is calculated in the same manner as equation (1), where it is considered effective because it is recorded during or at the end of a light period just before closing or changing the light intensity. LC3 pre-configured measurement was used with 7 steps increase in actinic light intensity within the PAM-fluorometer: 10, 20, 50, 100, 300, 500, 1000 µmol photons m^−2^ s^−1^. Parameterization of the light response curve followed Eilers and Peeters (1988) formulation [43]: A linear regression curve was fitted to the logarithmic data with the following fit equation:(2)$$p=\frac{I}{{\mathrm{aI}}^{2}+\mathrm{bI}+c}$$
where p and I are the photosynthetic activity and light intensity, respectively. The coefficients a, b and c are used to minimize the fit over the logarithmic data in order to extract the required information. Their initial values at the start of the fitting procedure are calculated directly from the acquired data:(3)$$a=\frac{1}{{\mathrm{sI}}_{m}^{2}}$$
(4)$$b=\frac{1}{P_{m}}-\frac{2}{{\mathrm{sI}}_{m}}$$
(5)$$c=\frac{1}{s}$$
where s, I_m_ and P_m_ are the initial linear slope, the light intensity at maximum photosynthetic activity, and the maximum photosynthetic activity reached. Then, after the fitting procedure is finished, reverse equations are used to extract the information needed:(6)$$\mathrm{LUE}=\frac{1}{s}$$
(7)$$\mathrm{Effective}\;\mathrm{Antenna}\;\mathrm{Size}\left(I_{k}\right)=\frac{c}{b+2\sqrt{\mathrm{ac}}}$$
(8)$$\mathrm{Maximum}\;\mathrm{activity}=\frac{1}{b+2\sqrt{\mathrm{ac}}}$$Light energy utilization distribution was calculated as suggested by Kramer et al. (2004) [29] where the puddle model [44] was taken into account during calculation:
(9)$$\Phi_{\mathrm{II}}=\frac{{F^{\prime }}_{m}-{F^{\prime }}_{s}}{{F^{\prime }}_{m}}$$
(10)$$\mathrm{NPQ}=\frac{F_{m}-{F^{\prime }}_{m}}{{F^{\prime }}_{m}}$$
(11)$$\Phi_{\mathrm{NO}}=\frac{1}{\mathrm{NPQ}+1+\Phi_{\mathrm{II}}\left(F_{m}/F_{0}-1\right)}$$
(12)$$\Phi_{\mathrm{NPQ}}=1-\Phi_{\mathrm{II}}-\Phi_{\mathrm{NO}}$$ where Φ_II_ is the effective quantum yield of PSII, F′_m_ and F′_s_ are the maximum and steady state fluorescence measured during light, respectively; NPQ initials stand for Non-Photochemical Quenching, Φ_NO_ represents the quantum yield of non-regulated loss in PSII, and Φ_NPQ_ is the quantum yield of regulated non-photochemical energy loss in PSII.Relaxation kinetics [45] were calculated during an induction-relaxation pre-configured measurement (NPQ1) in the portable fluorometer. The induction period was given at 60 s at an intensity of 50 µmol photons m^−2^ s^−1^ (comparable to the ambient light intensity the leaves experienced during acclimation in the growth chamber). Then, the relaxation period after illumination in the dark was given at 88 s, during which three saturating pulses are fired at a pre-set intervals of 26 s each in the dark. The calculations for each of the coefficients are:
(13)$$\mathrm{qE}=\frac{{F^{\prime }}_{m}^{2}-{F^{\prime }}_{m}^{1}}{F_{m}-F_{0}}$$
(14)$$\mathrm{qT}=\frac{{F^{\prime }}_{m}^{3}-{F^{\prime }}_{m}^{2}}{F_{m}-F_{0}}$$
(15)$$\mathrm{qI}=\frac{F_{m}-{F^{\prime }}_{m}^{3}}{F_{m}-F_{0}}$$ where F′^#^m is the number of the saturating pulse during the relaxation part of the induction- relaxation protocol.

### 4.3. The 77K Fluorescence Assay

Samples of apple tree leaves sprayed with 0.6% pure Metamitron (Sigma-aldrich, Rehovot, Israel), 0.096% DCMU (Sigma-aldrich, Israel), and DDW, were cut two days after administration and kept in liquid nitrogen until measurement. Then, each leaf was crushed with a mortar and pestle with liquid nitrogen and thawed in 1 mL PBS (1:10 with DDW, Sigma-aldrich, Israel). Immediately after thawing, a sample was loaded into a Pasteur pipette and was mounted into a Dewar cuvette filled with liquid nitrogen in spectrofluorometer (Cary-Varian, Agilent, Santa Clara, CA, USA). Preliminary tests were performed on the leaves in order to determine the volume of buffer and, therefore, the correct concentration used, which eliminates self-re-absorption of the fluorescence signal by the sample. Finally, each spectrum was pre-processed before comparison, and the data was normalized to the 685 nm peak and shifted on the *Y*-axis to zero by the first point in the curve (662 nm).

### 4.4. SDS-PAGE and Western Analysis

Apple tree leaves were homogenized with a mortar and pestle in the dark at 4 °C within liquid nitrogen, and then proteins were extracted with a lysis buffer (20% SDS, 1.5 M Tris pH 8.8), 1 mM PMSF and 1% phosphatase inhibitor (p0044, Sigma-aldrich). Samples were centrifuged at 4 °C, 14,000 RPM for 10 min and total protein was assessed with Bradford assay [46]. A total of 25 μgr total proteins were loaded in each well onto an SDS-PAGE [47] and transferred to a PVDF membrane (Amersham Biosciences, Amersham, UK), blocked for two hours in 5% (*v/v*) skim milk, and incubated with primary antibodies: De-phosphorylated Psb-a (1:10,000) (AS05-084, Sigma-aldrich), phosphorylated Psb-a (1:10,000) (AS13-2669, Agrisera) or Psb-o (AS05-092, Agrisera) (1:5,000) diluted according to the manufacturer’s recommendations. The membrane was then incubated with a goat α-rabbit –HRP conjugated secondary antibody (1:20,000) (AS09-602, Agrisera) and developed with an ECL-bright (AS16-ECL-N-10, Agrisera).

### 4.5. Statistical Analysis

Each group of control or treatments within this study included 3 samples from each of the three branches taken from different trees in the orchard. Statistical procedures were performed as described in Liran et al. (2020) [48]. Groups were checked for normal distribution with Shapiro–Wilk’s test and homogeneity of variances with Levene’s test. If both tests were satisfied, analysis of variance (ANOVA) was selected. Pairwise comparisons were checked with Tukey’s HSD test. If the homogeneity of variance test was violated, Welch’s ANOVA was used instead and Games–Howell (for epsilon < 0.75) or Greenhouse–Geisser (for epsilon > 0.75) post hoc comparison tests were used. If the normality criteria were violated, a Kruskal–Wallis’s a-parametric ANOVA was used with Dunn’s procedure and a Bonferroni adjustment were made for the pairwise comparisons.

For inspection of differences between groups in time, a repeated measures ANOVA was used if Mauchly’s sphericity test was satisfied. If homoscedasticity was not violated, pair-wise comparisons were checked with Pearson’s correlation tests and Bonferroni adjustments. If homoscedasticity test failed, a Spearman’s rank non-parametric correlation test was used instead. If Mauchly’s sphericity test was violated, a Friedman’s a-parametric ANOVA was applied with pair-wise comparisons with Bonferroni’s adjustments. Statistical significance was set to *p* < 0.05. Statistical analyses were carried out in Statistical Product and Service Solution (SPSS) (IBM, Chicago, IL, USA).

## Figures and Tables

**Figure 1 plants-10-02803-f001:**
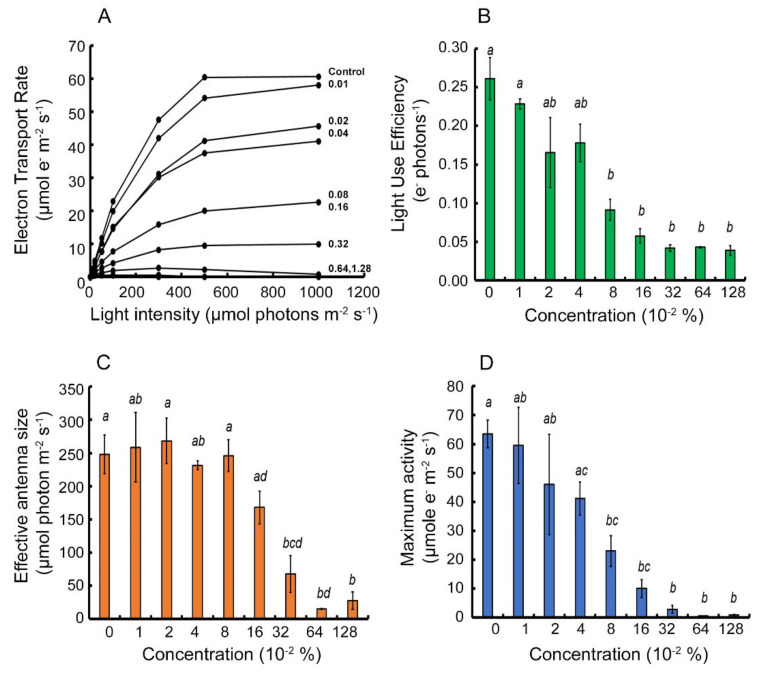
Parameterization of light response curves of Apple (*Malus domestica*) cultivar “Golden Delicious” that experienced a Brevis^®^ inhibitor gradient for 2 days under laboratory conditions. Panel (**A**) represents light response curves data. Panels (**B**–**D**) represent the parametrization variables-Light Use Efficiency (LUE) (green), Effective antenna size (orange), and maximum Electron Transport Rate (ETR) (blue), respectively. Each column is an average of three biological repeats. Error bars represent standard error of the mean. Amount of inhibitor is given in percentage. Letter notations describe statistically significant difference at *p* < 0.05.

**Figure 2 plants-10-02803-f002:**
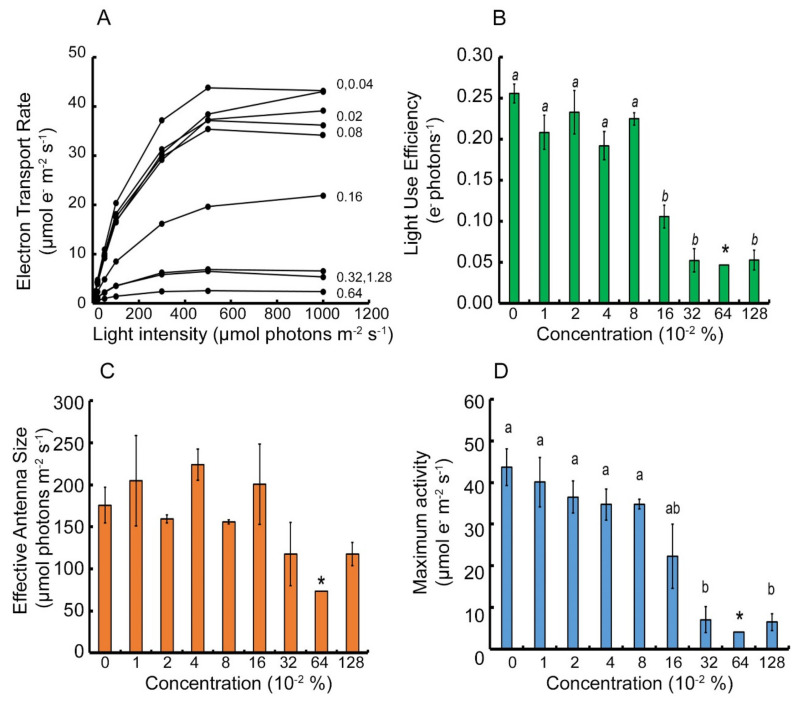
Parameterization of light response curves of Apple (*Malus domestica*) cultivar “Top Red” that experienced a Brevis^®^ inhibitor gradient for 2 days under laboratory conditions. Panel (**A**) represents light response curves data. Panels (**B**–**D**) represent the parametrization variables-Light Use Efficiency (LUE) (green), Effective antenna size (orange), and maximum Electron Transport Rate (ETR) (blue), respectively. The asterisk in the panels represent two biological repeats only, therefore no error bars are present. Each column is an average of three biological repeats. Error bars represent standard error of the mean. Inhibitor is given in percentage. Letter notations describe statistically significant difference at *p* < 0.05.

**Figure 3 plants-10-02803-f003:**
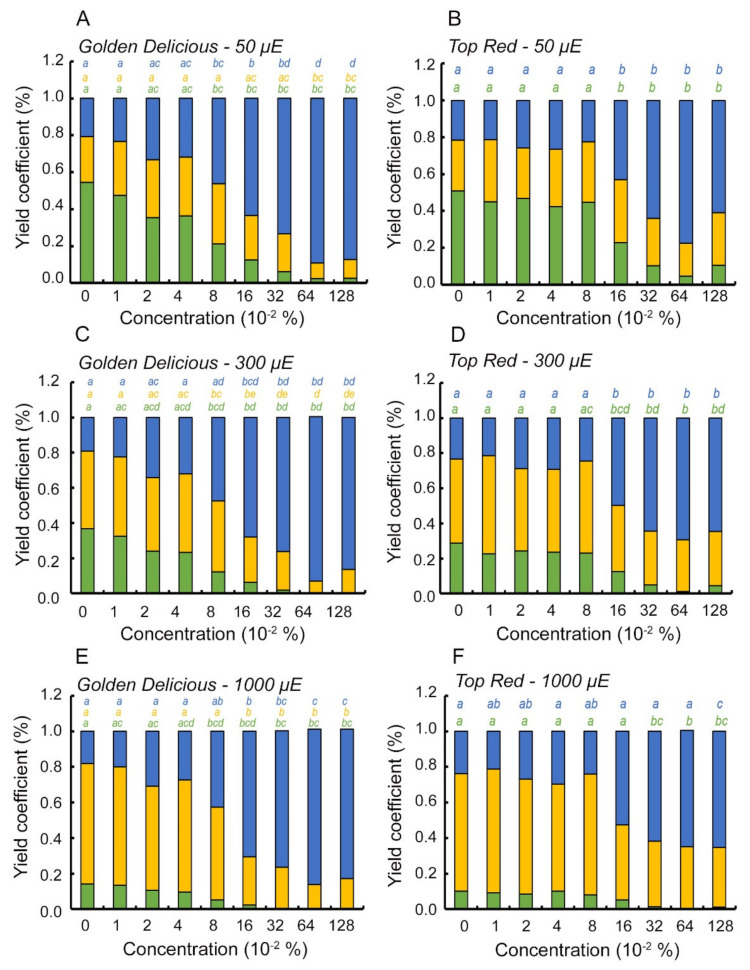
Quantum yields of absorbed energy distribution between the three outcomes of light energy utilization in two apple cultivars with Brevis^®^ inhibitor. Panels (**A**,**C**,**E**) and (**B**,**D**,**F**) represent results of light energy utilization distribution analysis across three light intensities extracted from the light response curve, for Golden Delicious and Top Red cultivars, respectively. Each column comprises three stacked bars which are summed to 1. The colors green, yellow and blue represent effective quantum yield of PSII photochemistry (Φ_PSII_), quantum yield of regulated non-photochemical energy loss in PSII (Φ_NPQ_) and quantum yield of non-regulated energy loss in PSII (Φ_NO_), respectively. Each bar represents three biological repeats. Letter notations describe statistically significant differences (*p* < 0.05). µE is µmol photon m^−2^ s^−1^.

**Figure 4 plants-10-02803-f004:**
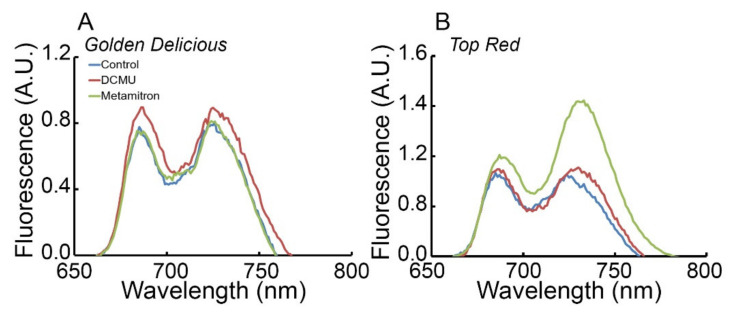
The 77K fluorescence spectra of apple tree leaves of two cultivars experiencing photosynthetic inhibition by Metamitron. Panels (**A**,**B**) represent Golden Delicious and Top Red cultivars, respectively. There are three treatments in each panel-control (only buffer), 3-(3,4-dichlorophenyl)-1,1-dimethylurea (DCMU), 4-Amino-4,5-dihydro-3-methyl-6-phenyl-1,2,4-triazin-5-one (Metamitron) in colors of Blue, Magenta and Green, respectively. Each curve is an average of three biological repeats.

**Figure 5 plants-10-02803-f005:**
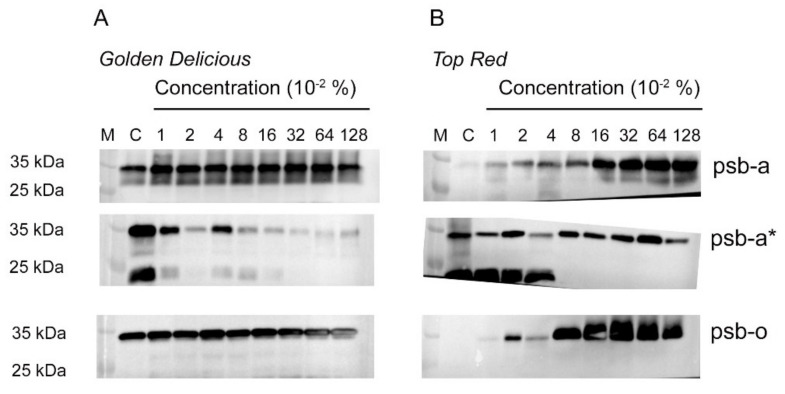
Western blot of three subunits of PSII for two apple cultivars exposed to a concentration gradient of the Brevis^®^ inhibitor. Panels (**A**,**B**) represent information for Golden Delicious and Top Red cultivars, respectively. Each sample includes three different leaf extracts. Psb-a, psb-a* and psb-o stand for D1, D1-phosphorylated and OEC-33 (Oxygen Evolving Center) subunits of PSII. M, C are Marker and Control (without Brevis), respectively.

**Figure 6 plants-10-02803-f006:**
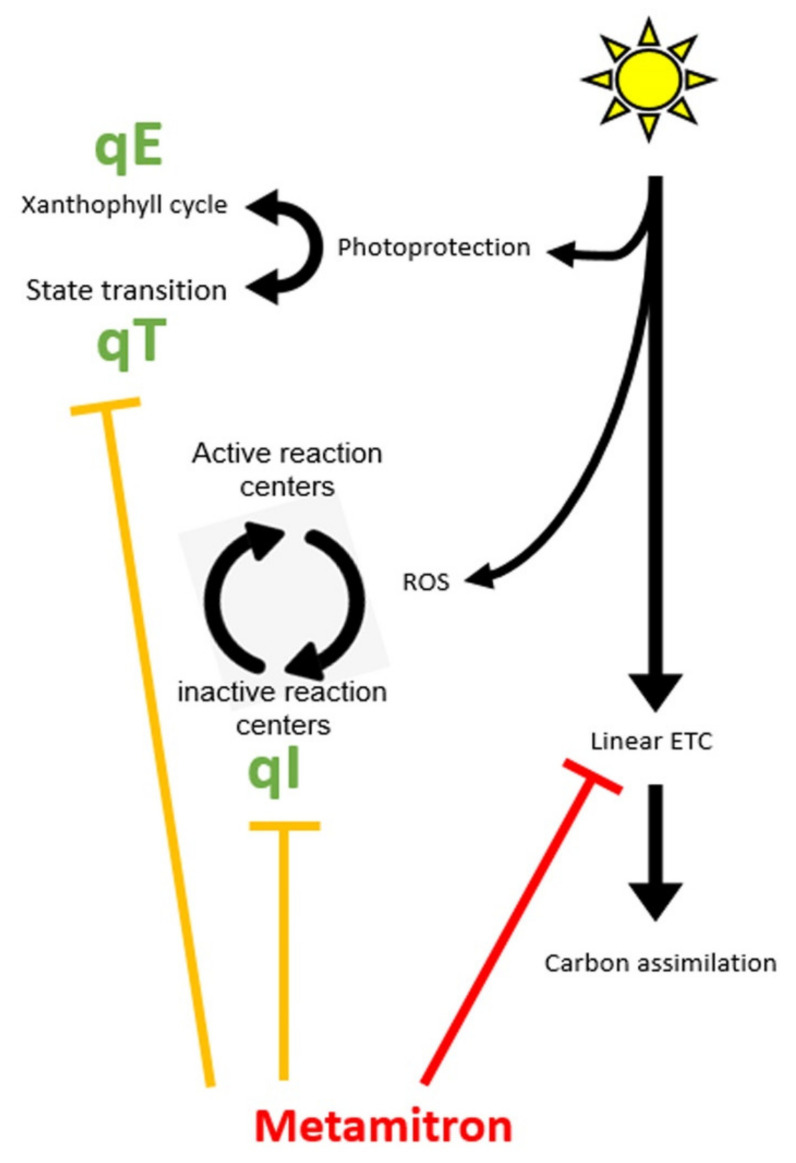
Metamitron activity model in view of this study. ETC refers to Electron Transport Chain. The bold red line represents the Metamitron primary mode of action blocking electron transport rate, and the bold orange lines represent its secondary activity, as revealed in this study.

**Table 1 plants-10-02803-t001:** Quantum yield of the photoprotective pathways of apple cultivar “Golden Delicious” on the background of Brevis^®^ inhibitor gradient and over time. Each value is an average of three biological repeats performed in duplicates, and the error values are standard error of the mean. Letter notations describe statistically significant difference (*p* < 0.05).

Days	Concentration (%)	Quantum Yield of Process ^‡^		
		qE	qT	qI
2	0	0.156 ± 0.090 ^ab^	0.096 ± 0.012 ^a^	0.536 ± 0.120 ^ab^
	0.01	0.143 ± 0.032 ^a^	0.136 ± 0.011 ^ab^	0.473 ± 0.049 ^a^
	0.05	0.336 ± 0.016 ^b^	0.166 ± 0.005 ^b^	0.257 ± 0.023 ^b^
	0.5	0.227 ± 0.027 ^ab^	0.154 ± 0.011 ^b^	0.411± 0.039 ^ab^
5	0	0.278 ± 0.081	0.149 ± 0.045	0.520 ± 0.186
	0.01	0.134 ± 0.076	0.105 ± 0.023	0.527 ± 0.132
	0.05	0.223 ± 0.022	0.171 ± 0.024	0.439 ± 0.076
	0.5	0.255 ± 0.045	0.189 ± 0.010	0.389 ± 0.068
7	0	0.114 ± 0.026	0.132 ± 0.027	0.710 ± 0.245
	0.01	0.063 ± 0.033	0.094 ± 0.034	0.521 ± 0.080
	0.05	0.168 ± 0.046	0.176 ± 0.033	0.524 ± 0.093
	0.5	0.152 ± 0.028	0.176 ± 0.014	0.498 ± 0.008
10	0	0.087 ± 0.012	0.129 ± 0.017	0.471 ± 0.030
	0.01	0.121 ± 0.020	0.127 ± 0.024	0.484 ± 0.038
	0.05	0.161 ± 0.020	0.173 ± 0.007	0.439 ± 0.022
	0.5	0.166 ± 0.033	0.170 ± 0.08	0.466 ± 0.042

^‡^ qE-Energy dependent quenching, qT-State transition dependent quenching, qI-photoinhibition dependent quenching.

**Table 2 plants-10-02803-t002:** Quantum yield of the photoprotective pathways of apple trees “Top Red” on the background of Brevis^®^ inhibitor gradient with time. Each value is an average of three biological repeats performed in duplicates, and the error values are standard error of the mean. Letter notation describe statistically significant difference at *p* < 0.05.

Days	Concentration (%)	Quantum Yield of Process ^‡^		
		qE	qT	qI
2	0	0.165 ± 0.003	0.140 ± 0.009	0.465 ± 0.028 ^a^
	0.01	0.194 ± 0.030	0.135 ± 0.014	0.430 ± 0.048 ^a^
	0.05	0.198 ± 0.022	0.138 ± 0.004	0.437 ± 0.028 ^a^
	0.5	0.241 ± 0.061	0.099 ± 0.048	0.285 ± 0.069 ^b^
5	0	0.149 ± 0.006	0.148 ± 0.017 ^a^	0.471 ± 0.044 ^a^
	0.01	0.123 ± 0.006	0.133 ± 0.004 ^a^	0.500 ± 0.002 ^a^
	0.05	0.143 ± 0.015	0.130 ± 0.012 ^a^	0.535 ± 0.036 ^a^
	0.5	0.199 ± 0.060	0.067 ± 0.020 ^b^	0.243 ± 0.015 ^b^
7	0	0.145 ± 0.018	0.154 ± 0.014	0.455 ± 0.032
	0.01	0.116 ± 0.040	0.122 ± 0.017	0.528 ± 0.073
	0.05	0.151 ± 0.025	0.160 ± 0.022	0.439 ± 0.060
	0.5	0.263 ± 0.087	0.133 ± 0.042	0.509 ± 0.133
10	0	0.040 ± 0.018	0.059 ± 0.019	0.534 ± 0.029 ^a^
	0.01	0.062 ± 0.018	0.089 ± 0.008	0.581 ± 0.005 ^a^
	0.05	0.045 ± 0.041	0.078 ± 0.033	0.626 ± 0.038 ^a^
	0.5	0.039 ± 0.022	0.033 ± 0.011	1.046 ± 0.121 ^b^

^‡^ qE-Energy dependent quenching, qT-State transition dependent quenching, qI-Photoinhibition dependent quenching.

## Data Availability

Data is available upon request from the corresponding author.

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
