# Peer review of "Metamitron, a Photosynthetic Electron Transport Chain Inhibitor, Modulates the Photoprotective Mechanism of Apple Trees"

_plants, 2021, doi:10.3390/plants10122803_

Round 1

Reviewer 1 Report

A brief summary

The aim of this work was to analyze the effect of Brevis on fruit thinning of two apple cultivars: Golden Delicious and Top Red. The correlation between photosynthesis light reaction of apple trees and metamitron was studied. Brevis is an innovative product that mimics shading by interfering with the photosynthetic process, curbing the production of carbohydrates used for tree growth. I found several articles about thinning efficacy of Brevis on different apple cultivars like Gala, Golden Delicious, Red Delicious or Granny Smith. However, I did not find any study on Top Red. In addition, this manuscript describes very well metamitron activity model. Therefore, the topic of presented study is relevant and describes current problem.

Broad comments

  1. I found the paper to be overall important and describing a very interesting findings. The experiment is designed properly. Introduction, Material and Methods, Results and Discussion are well described. Abstract has a lot of misspellings and needs to be edited and revised.
  2. The manuscript is well written, clear, precise, and easy to read but it requires some editing. Please, see specific comments. English should be revised too.
  3. Variety/cultivar names - Golden Delicious and Top Red should start with a capital letter. Please, revise all the manuscript.
  4. I suggest adding Conclusions or a couple of sentences in Discussion section about a practical applications of the results and future work.

Specific comments

Line 132, 168 Malus domestica instead of Melus x Domestica

Line 213 two apple cultivars instead of two apple tree cultivars

Line 232, 241 Please, revise those paragraphs. New paragraph should start with a new sentence.

Author Response

Dear Reviewer, 

Thank you very much for your comments and suggestions. Below, please find a response in color per every section. We received all your comments and added text where needed in the manuscript.

Broad comments

  1. I found the paper to be overall important and describing a very interesting findings. The experiment is designed properly. Introduction, Material and Methods, Results and Discussion are well described. Abstract has a lot of misspellings and needs to be edited and revised. Thank you very much. The Abstract was revised and edited as requested by a native American speaker editor.
  2. The manuscript is well written, clear, precise, and easy to read but it requires some editing. Please, see specific comments. English should be revised too. The Manuscript was revised and edited as requested. 
  3. Variety/cultivar names - Golden Delicious and Top Red should start with a capital letter. Please, revise all the manuscript. All the manuscript was revised for this correction including the initials.
  4. I suggest adding Conclusions or a couple of sentences in Discussion section about a practical applications of the results and future work.  A future work suggestions and possible implications of this study have been brought forth and added in lines 460-472.

Specific comments

Line 132, 168 Malus domestica instead of Melus x Domestica – Corrected as requested

Line 213 two apple cultivars instead of two apple tree cultivars - Corrected as requested, and through manuscript.

Line 232, 241 Please, revise those paragraphs. New paragraph should start with a new sentence. – Corrected as requested.

Reviewer 2 Report

The study by Tadmor and coauthors examines the effects of a commercial thinner on apple leaves PSII photochemistry, photoprotective mechanisms and ROS damage evidenced by chlorophyll fluorescence and PSII protein analysis.

The experimental part is solid and straightforward. The conclusions are based on experimental evidence (with the exceptions included in comment 1) and the presentation is quite well prepared. Some details are missing and also, some parts of the text need some revising.

Major issues:

1) There is no mechanistic explanation of the effects of metamitron on qT and qI but only on ETR based on the inhibition of PQ reduction. NPQ effects could be indirect due to the suppression of the electron transport. Off course there is a striking difference between metamitron and diuron based on the 77 °K fluorescence spectra but this accounts only for the qT component and it is also an evidence of its effect and not a mechanistic explanation.

This is not a problem of the study but this gap in our knowledge should be included in the discussion as a future outlook.

2) While there is a reduction of NPQ in higher metamitron concentrations (e.g. Fig. 4E), there is no such reduction in qE+qT+qI (or qE alone as an NPQ analogue) along the concentration gradient at 2 days after the application (Table 1). Is there an explanation for this discrepancy?

3) Figs 1 and 7 are redundant at some extend and could be merged to a single figure with a colour coding referring to what is known before and after the present study.

4) More details are needed on the chlorophyll PAM fluorescence protocols. Details in the Materials and Methods do not allow the repetition of the experiments.

5) Figure legends need a text polishing. Statistical annotations should be explained in the legend text.

6) The text needs a good revision. For instance, qI is referred as inhibition/damage, downregulated PSII and repair cycle. Metamitron is called also thinner and inhibitor. There are some more similar issues, e.g. ‘downstream PSII’ is different that ‘downstream to PSII’.

Some minor corrections are given in the uploaded pdf copy.

Author Response

Dear Reviewer, 

Thank you very much for your comments and suggestions. We received all the comments and corrected the manuscript accordingly. Below, please find a point by point answer and a reference to the respected lines within the manuscript. All the responses are colored, both here and within the manuscript. Special thanks for correcting the English. It was also sent to a professional native American speaker editor.

Major issues:

1) There is no mechanistic explanation of the effects of metamitron on qT and qI but only on ETR based on the inhibition of PQ reduction. NPQ effects could be indirect due to the suppression of the electron transport. Off course there is a striking difference between metamitron and diuron based on the 77 °K fluorescence spectra but this accounts only for the qT component and it is also an evidence of its effect and not a mechanistic explanation.

This is not a problem of the study but this gap in our knowledge should be included in the discussion as a future outlook.  – Added a paragraph as a closing argument for the discussion in lines 473 -481 with a future outlook for our suggestion of what should be done in order to establish a mechanistic model.

2) While there is a reduction of NPQ in higher metamitron concentrations (e.g. Fig. 4E), there is no such reduction in qE+qT+qI (or qE alone as an NPQ analogue) along the concentration gradient at 2 days after the application (Table 1). Is there an explanation for this discrepancy? – a limitation paragraph with this discrepancy is added in lines 443-451 in the discussion. A possible explanation is suggested by us as an interpretation of these results.

3) Figs 1 and 7 are redundant at some extend and could be merged to a single figure with a colour coding referring to what is known before and after the present study.  –Figure 1 was removed and the new Figure 6 was corrected according to the reviewer suggestion, with legend text corrected as well in L. 426-428.

4) More details are needed on the chlorophyll PAM fluorescence protocols. Details in the Materials and Methods do not allow the repetition of the experiments.  – Materials and methods section was added regarding each of the protocols used to analyze chlorophyll a fluorescence in lines 497-579, including all the equations used in the process.

5) Figure legends need a text polishing. Statistical annotations should be explained in the legend text. – statistical annotation explanation added to each figure where needed, text was slightly polished with the aid of the professional editor.

6) The text needs a good revision. For instance, qI is referred as inhibition/damage, downregulated PSII and repair cycle. Metamitron is called also thinner and inhibitor. There are some more similar issues, e.g. ‘downstream PSII’ is different that ‘downstream to PSII’. We took a great care to fix all the redundant terms. We couldn't change all the terms for Metamitron (specifically in the Abstract). Because it is used as a thinner, but its main activity is inhibition.

Some minor corrections are given in the uploaded pdf copy. All the corrections suggested were performed. The authors thank the reviewer for his dedicated assistance with the English.